# Evidence-Based Management of Burns: A Narrative Review of Evolving Practices

**DOI:** 10.3390/ebj6040059

**Published:** 2025-11-10

**Authors:** Anna Jolly Neriamparambil, Raja Sawhney, Wei Lun Wong

**Affiliations:** 1Department of Plastic and Reconstructive Surgery, Gold Coast University Hospital, 1 Hospital Boulevard, Southport, Gold Coast, QLD 4215, Australia; raja.sawhney@health.qld.gov.au; 2School of Medicine, The University of Queensland, Sir Fred Schonell Drive, St. Lucia, QLD 4071, Australia; 3National Burn Centre, Department of Plastic and Reconstructive Surgery, Middlemore Hospital, Auckland 2025, New Zealand; wongweilun@gmail.com; 4Department of Surgery, The University of Auckland, 34 Princes Street, Auckland Central, Auckland 1010, New Zealand

**Keywords:** burn management, enzymatic debridement, scar, metabolic modulation, dermal substitute, keratinocyte spray

## Abstract

**Background:** The last decade has seen transformative changes in burn care, driven by advances in pharmacology, regenerative medicine, surgical techniques, and digital technologies. As management strategies evolve beyond survival to encompass functional and esthetic recovery, this review consolidates current evidence to inform best practice. **Methods:** A comprehensive narrative review was conducted using PubMed to identify peer-reviewed English-language articles from the past 10 years relevant to acute and long-term burn management. Selection focused on high-level evidence, including randomized controlled trials, systematic reviews, and meta-analyses, emphasizing novel and evolving clinical interventions. **Results:** Key advances include the integration of propranolol and oxandrolone for metabolic modulation; enzymatic debridement agents such as NexoBrid^®^; regenerative approaches like epidermal cell sprays (e.g., RECELL^®^) and dermal substitutes (e.g., Integra^®^, MatriDerm^®^, NovoSorb^®^ BTM); and innovations in scar modulation, notably fractional CO_2_ laser therapy. The emergence of 3D bioprinting, and artificial intelligence further supports a shift toward precision burn medicine. **Conclusions:** Burn management is evolving from protocol-driven to patient-centred care, underpinned by high-quality evidence and technological innovation. The integration of systemic, local, and rehabilitative strategies is improving outcomes in survival, function, and quality of life. Ongoing challenges include cost, access, and translation of novel therapies into widespread clinical practice.

## 1. Introduction

Burn injuries remain a significant public health concern worldwide, contributing substantially to morbidity, mortality, and healthcare resource utilization [1]. Globally, an estimated 11 million people require medical attention for burn injuries each year, with over 180,000 deaths, predominantly in low- and middle-income countries [1]. Despite improvements in prevention and acute care, burn injuries continue to present complex challenges due to the multisystemic physiological responses they provoke, the need for prolonged rehabilitation, and the risk of lifelong functional and esthetic sequelae. As such, optimizing burn care remains a dynamic area of clinical and translational research.

Historically, the management of burns has been anchored in early excision and grafting, fluid resuscitation, infection control, and nutritional support. While these foundational principles remain vital, the last decade has witnessed a paradigm shift towards more individualized, evidence-based, and biologically driven approaches [2]. Innovations in pharmacotherapy, tissue engineering, metabolic support, enzymatic debridement, and scar modulation are now redefining standards of care. These advances aim not only to improve survival but also to enhance quality of life, minimize donor site morbidity, accelerate healing, and reduce the burden of long-term scarring and contractures [2].

Among the most noteworthy developments are pharmacological agents that target the hypermetabolic and catabolic responses triggered by major burns. The anabolic steroid oxandrolone and the β-adrenergic blocker propranolol have emerged as important adjuncts, showing consistent benefits in preserving lean body mass, improving wound healing, and attenuating the prolonged stress response [3,4]. Meanwhile, metabolic monitoring through indirect calorimetry has allowed for more precise nutritional interventions, supporting recovery and immune function in critically ill burn patients [4].

At the local wound level, the landscape of debridement has shifted with the adoption of enzymatic agents such as bromelain-based formulations (e.g., NexoBrid^®^), which offer targeted, non-surgical eschar removal [5]. These agents provide an alternative to traditional tangential excision, potentially preserving viable dermis, reducing blood loss, and expediting wound closure. In parallel, regenerative technologies have flourished. Epidermal cell sprays, such as the RECELL^®^ system, and cultured skin substitutes are enabling coverage of extensive wounds with smaller donor site requirements, thereby reducing patient morbidity [6].

Dermal regeneration has also evolved, with acellular matrices like Integra^®^, MatriDerm^®^ and NovoSorb^®^ BTM offering scaffolds that support neodermis formation, particularly valuable in deep burns and areas of high functional or cosmetic importance [7]. Additionally, advances in scar management, notably through fractional CO_2_ laser therapy, are allowing for the modulation of hypertrophic and restrictive scars with improved outcomes in pliability, texture, and pigmentation [8].

This narrative review aims to synthesize the high-quality evidence published over the past 10 years to provide clinicians and researchers with a comprehensive understanding of contemporary, evidence-based approaches to burn care. It will critically evaluate the clinical efficacy, safety, and practical applications of emerging therapies, while also considering the limitations and future directions in this rapidly evolving field. Emphasis is placed on interventions that not only improve survival but also promote holistic recovery—functional, psychological, and esthetic—aligning with the modern ethos of patient-centred care in reconstructive surgery. While some of these treatments have been around for longer, emerging evidence and practicality have resulted in more recent adoption into clinical practice.

## 2. Materials and Methods

A comprehensive literature search was conducted to identify contemporary therapeutic strategies in burn management. The PubMed database was queried using the search terms (burn therapy) OR (burn injury), resulting in approximately 127,600 records published between 1940 and 2023. To capture recent advances, the analysis was restricted to studies published during the most recent decade (December 2013–December 2023). From this subset, 48,137 records were screened based on titles and abstracts to identify publications of potential relevance.

Subsequent targeted searches were performed to address specific thematic areas relevant to burn care. The Boolean string employed was: (injury OR treatment) AND (burns) AND (escharotomy OR debridement OR topical treatment OR dressing OR excision OR skin substitutes OR autografts OR skin grafts OR cultured epithelial autograft OR xenografts OR allografts OR human amnion OR artificial skin OR maggot debridement therapy OR negative pressure wound therapy OR hyperbaric oxygen therapy OR fish skin grafts OR nutrition OR pain management OR psychological advisory OR rehabilitation OR scar treatment OR antibiotic therapy).

Eligible publications included original research articles, randomized controlled trials, clinical trials, case reports, and systematic or narrative reviews, all published in English. Only studies with full-text access were considered to ensure comprehensive data extraction and critical appraisal. Historically significant older articles were retained when they contributed contextually relevant insights. Duplicate entries were excluded prior to analysis. Eligibility for inclusion was independently assessed by the senior authors, who evaluated each study for novelty, methodological quality, and clinical relevance to current burn treatment practices. A total of 468 studies met the inclusion criteria and were subsequently categorized according to major therapeutic domains. The selection and screening process are illustrated in the accompanying PRISMA flow diagram in Figure 1.

## 3. Review

### 3.1. Pharmacological Adjuncts in Burn Recovery

Severe burn injuries trigger a profound and prolonged hypermetabolic and catabolic state, characterized by elevated catecholamines, cortisol, glucagon, and inflammatory cytokines. This stress response, often persisting for up to two years post-injury, contributes to muscle wasting, impaired wound healing, growth arrest (in children), insulin resistance, and increased morbidity [9,10]. Pharmacologic adjuncts have been developed to attenuate this response, preserve lean body mass, improve wound healing, and support recovery. Among these, oxandrolone and propranolol have demonstrated the most consistent clinical efficacy and have been integrated into modern burn protocols, particularly in specialized burn centres [4].

#### 3.1.1. Oxandrolone

Oxandrolone is a synthetic, orally bioavailable anabolic androgenic steroid (AAS), derived from dihydrotestosterone, with minimal virilizing effects. It exerts its effects through androgen receptor-mediated transcriptional activation, leading to increased protein synthesis and nitrogen retention, as well as inhibition of glucocorticoid-induced catabolism [3].

Over the last decade, numerous studies have reinforced the therapeutic benefits of oxandrolone in burn care. In pediatric and adult populations with burns exceeding 20% of the total body surface area (TBSA), oxandrolone has been associated with accelerated wound healing, decreased length of hospital stays, improved bone mineral density, and restoration of lean body mass [10]. Studies have demonstrated that when administered at 0.1 mg/kg twice daily in the pediatric cohort and 10mg twice daily in the adult cohort, oxandrolone significantly improves net protein balance and facilitates skin graft take and epithelialization [11].

A 2020 meta-analysis by Ring et al., evaluating randomized controlled trials (RCTs) and prospective cohort studies, concluded that oxandrolone is a beneficial adjunct to acute care of burn patients by shortening hospital stays and improving several growth and wound healing parameters, although asymptomatic elevations in liver enzymes were noted [3]. Importantly, oxandrolone’s anabolic effects also support recovery during rehabilitation by promoting musculoskeletal strength and function, essential for return to independence and work.

Despite its promising profile, oxandrolone use remains limited in some settings due to concerns regarding long-term androgenic effects, potential hepatotoxicity, and regulatory constraints. Continued surveillance for side effects and individualized dosing based on patient-specific factors remain essential [10]. In June 2023, the FDA formally withdrew approval for oxandrolone, making them unavailable in standard retail pharmacies but can still be legally obtained through patient-specific compounding with a valid prescription.

#### 3.1.2. Propranolol

Propranolol, a non-selective beta-adrenergic antagonist, represents a cornerstone in modulating the burn-induced catecholaminergic surge [4]. Elevated plasma catecholamine levels after burns lead to increased myocardial oxygen consumption, tachycardia, insulin resistance, lipolysis, and immune dysfunction. By attenuating β1- and β2-adrenergic receptor activity, propranolol reduces heart rate, cardiac work, and lipolysis, thereby conserving protein stores and modulating inflammation [10].

Multiple studies have demonstrated that propranolol, dosed to reduce heart rate by 15–20%, can significantly blunt the hypermetabolic state [12,13]. Randomized trials in pediatric populations showed improved lean body mass retention, decreased skin graft loss, and reduced resting energy expenditure [12]. Additionally, propranolol has been shown to preserve muscle protein synthesis and reduce burn-induced insulin resistance.

Data on treatment with propranolol in adult burn patients are not as extensive as in the pediatric cohort. It has been shown to improve wound healing, shorten length of hospital stays and decrease blood loss during skin grafting [10]. Propranolol has also been explored for its anti-inflammatory and anti-fibrotic effects, which may be relevant in minimizing hypertrophic scarring and post-burn fibrosis. Preclinical models suggest that propranolol can reduce the expression of TGF-β and other fibrogenic cytokines, offering an additional avenue of therapeutic benefit [4].

Clinical implementation of propranolol requires careful titration and cardiovascular monitoring, especially in patients with pre-existing bradyarrhythmias, hypotension, or respiratory compromise. Nonetheless, in appropriately selected patients, propranolol has emerged as a safe and effective adjunctive therapy up to one year post burn and is not associated with increased risk of sepsis [10].

#### 3.1.3. Chemical Venous Thromboembolism Prophylaxis

A meta-analysis of 9619 patients demonstrated that low molecular weight heparin is more effective than unfractionated heparin and mechanical compressive devices in reducing the incidence of deep venous thrombosis and should be considered the primary pharmacological agent for thromboprophylaxis in critically ill patients [14]. According to the Surviving Sepsis After Burn Campaign, chemical VTE prophylaxis should be commenced within 12 h of admission and there was no consensus as to the use of anti-factor Xa levels to adjust dosing [15].

#### 3.1.4. Tranexamic Acid

Burn injury causes coagulopathy that is likely multifactorial, partly attributed to haemodilution secondary to aggressive resuscitation to manage significant fluid losses. Significant bleeding is encountered, particularly during early management of these injuries with debridement and grafting. Tranexamic acid (TXA) is an anti-fibrinolytic pharmacological adjunct that reduces blood loss and transfusion requirement in burn surgery without increasing the risk of VTE events or mortality [16].

#### 3.1.5. Summary

Both oxandrolone and propranolol represent pivotal pharmacologic interventions that address the systemic consequences of major burns. Their ability to modulate the endocrine and metabolic environment offers substantial benefits beyond the wound itself, contributing to improved survival, faster rehabilitation, and better functional outcomes [3,13]. These agents exemplify the trend in modern burn care towards targeted, physiology-based therapies that support the body’s reparative processes while minimizing complications. Use of enoxaparin with anti-Xa monitoring reduces the incidence of VTE events [17]. Administration of TXA reduces blood loss and transfusion requirement without increasing VTE events or mortality [16]. Future research will likely refine indications, dosing protocols, and patient selection strategies, enhancing the integration of these agents into personalized burn treatment regimens.

### 3.2. Debridement Innovations

Timely and effective wound debridement remains a cornerstone of acute burn care [18]. Removal of necrotic eschar reduces bacterial burden, mitigates systemic inflammatory responses, facilitates accurate wound assessment, and prepares the wound bed for definitive closure [5,19]. Traditionally, this has been achieved through surgical excision—most commonly tangential or fascial excision—followed by autografting. While effective, surgical debridement is associated with significant blood loss, potential removal of viable tissue, and the need for general anesthesia and operative theatre access, all of which can delay intervention, particularly in resource-constrained or mass-casualty settings [20].

In response to these limitations, the past decade has seen the emergence and validation of enzymatic debridement agents, offering a non-surgical alternative for selective eschar removal [18]. Among these, bromelain-based enzymatic debridement, most notably NexoBrid^®^ (a concentrate of proteolytic enzymes derived from *Ananas comosus*), has garnered considerable attention and regulatory approval in multiple jurisdictions [5].

#### 3.2.1. Bromelain-Based Enzymatic Debridement

NexoBrid^®^ represents a shift towards minimally invasive burn wound care. Applied topically to thermal burns (usually within 72 h of injury), it selectively degrades necrotic tissue while preserving viable dermis and adnexal structures. Its mechanism of action relies on the targeted proteolytic cleavage of denatured proteins within the eschar, without systemic absorption or immunogenicity [5].

Several large-scale clinical trials and post-marketing studies over the past decade have established the efficacy and safety of bromelain-based debridement [20,21]. Systematic analysis of randomized controlled trials demonstrated that NexoBrid^®^ accelerated eschar removal and reduced the need for surgical excision and autografts, without adversely affecting wound closure tome or long-term scar quality [21].

Moreover, enzymatic debridement is best indicated for mid to deep dermal burns with mixed patterns, but can be applied in full thickness burns and might be less effective in scald injuries or chemical burns [5]. Early application can prevent burn related compartment syndrome in circumferential extremity or extensive trunk burns, but cannot replace surgical release in case of established respiratory compromise [5]. Enzymatic debridement has shown a favourable impact on long-term functional and esthetic outcomes. By preserving dermal structures and minimizing collateral damage, it supports spontaneous epithelialisation in intermediate-depth burns, reducing the need for grafting and thereby minimizing donor site morbidity. Several studies have reported improved scar quality, range of motion, and patient-reported satisfaction at 6 to 12 months post-treatment [21].

From a practical standpoint, NexoBrid^®^ allows for bedside application, which can be especially advantageous in pediatric, elderly, or medically unstable patients who may not tolerate surgery. It is also being explored as a field-deployable option in disaster and military burn care scenarios [18].

However, enzymatic debridement is not without limitations. The procedure can cause significant pain during the active debridement phase, necessitating effective analgesia and sometimes sedation. Furthermore, while systemic adverse events are rare, there is a theoretical risk of bleeding or secondary infection if used inappropriately on highly vascularized areas or in immunocompromised hosts. Cost and clinician familiarity also remain barriers to widespread adoption, particularly outside tertiary burn centres [18,20].

In 2025, a comprehensive meta-analysis by De Freitas et al. examined 7 studies comprising 484 patients of whom 238 (49%) were treated with enzymatic debridement and concluded that enzymatic debridement with bromelain presented a significant reduction in need for surgical excision (RR 0.17) and need for autografts (RR 0.40) without any difference in time to wound closure or Modified Vancouver Scar Scale [21]. These findings support its inclusion as a first-line or adjunctive modality in selected patients.

#### 3.2.2. Summary

Enzymatic debridement with bromelain-based agents represents a major advancement in the non-surgical management of burns. By offering a selective, effective, and tissue-sparing approach to eschar removal, it aligns with contemporary goals of minimizing invasiveness, accelerating wound closure, and enhancing long-term outcomes [5]. Its role is likely to expand further as clinician familiarity grows and as adjunctive technologies evolve to complement this novel approach to burn wound bed preparation.

### 3.3. Skin Replacement and Regenerative Technologies

Burn injuries involving full-thickness and deep partial-thickness skin loss often require prompt and durable wound closure to prevent infection, modulate the inflammatory response, and minimize scarring and contractures. Autologous split-thickness skin grafting (STSG) remains the gold standard. Fibrin sealants demonstrate a trend towards lower rates of hematoma or seroma and may be a better choice for adherence of skin grafts in topographically complex regions [22]. In patients with limited donor sites, micrografting (MEEK, Xpansion, or Pixel) techniques are utilized [23]. However, these options are associated with donor site morbidity, limited availability in large burns, and suboptimal esthetic or functional results in certain anatomical regions. These limitations have driven innovation in regenerative skin substitutes and cell-based therapies, aimed at accelerating wound coverage while preserving or restoring skin architecture and function [19,24].

#### 3.3.1. Epidermal Cell Sprays

Autologous epidermal cell spray systems represent a significant advancement in the field of burn care [6]. The most widely studied platform, RECELL^®^ (Avita Medical), enables point-of-care harvesting of a small split-thickness skin biopsy (approximately 1.5 cm^2^), which is enzymatically disaggregated to create a suspension of keratinocytes, melanocytes, Langerhans cells, and basal epidermal stem cells. This suspension is then sprayed onto the wound bed, promoting re-epithelialization and pigmentation over a larger surface area than traditional grafts [25,26].

Several randomized controlled trials have established RECELL’s efficacy as a primary modality in superficial or deep partial thickness burns and as an adjunct in full-thickness burns, demonstrating comparable wound closure rates to meshed STSGs with a significantly smaller donor site requirement [26]. Patients treated with RECELL plus STSG required up to 80% less autologous skin grafts compared to conventional meshed STSG alone, with similar re-epithelialization rates and superior donor site healing [25,26].

RECELL is particularly valuable when donor sites are limited (e.g., in large TBSA burns or pediatric patients), and has shown benefits in minimizing donor site pain, scarring, and procedural time. In addition, its ability to deliver melanocytes has demonstrated promise in restoring pigmentation in facial and extremity burns, with better colour match and reduced dyschromia compared to STSGs [25].

Despite its advantages, RECELL is typically used for smaller areas and in combination with STSGs in full-thickness wounds, as it lacks the dermal component required for durable coverage and biomechanical strength. Other concurrent developments include the CellMist Solution and SkinGun device (RenovaCare, New York, NY, USA) and SkinTE (PolarityTE, Salt Lake City, UT, USA). There are no comprehensive studies evaluating cost effectiveness of epidermal cell sprays [6,25,26].

#### 3.3.2. Dermal Substitutes

The past decade has seen an expansion in the clinical application of dermal regeneration templates, such as Integra^®^, MatriDerm^®^, and NovoSorb^®^ Biodegradable Temporizing Matrix (BTM) which provide a scaffold for neodermis formation prior to or concurrent with epidermal resurfacing. These acellular matrices are composed of bovine or porcine collagen, sometimes cross-linked with glycosaminoglycans, and are used either as temporary wound coverage or in two-stage reconstructions [27].

A 2024 meta-analysis of 31 comparative trials by Van Den Bosch et al. concluded that dermal substitutes reduce hypertrophic scarring, improve graft take, and result in superior esthetic and functional outcomes, especially in high-mobility or cosmetically sensitive areas (e.g., hands, neck, and face) [28]. Integra^®^, in particular, has been extensively validated in full-thickness burns requiring excision, where it is applied to the debrided wound bed and allowed to vascularize over 14–21 days, followed by application of an ultrathin autograft [29].

Single-stage reconstruction using MatriDerm^®^ and STSG has also gained traction, with studies reporting excellent pliability, reduced contraction, and improved elasticity. Randomized controlled trials support the use of MatriDerm in acute burns and contracture releases, with particular benefit in reducing donor site morbidity and surgical complexity. The integration of dermal substitutes with adjunctive therapies, such as negative pressure wound therapy (NPWT), has further enhanced graft take and angiogenesis, representing a synergistic approach to complex wound management [28,29].

BTM is a versatile dermal template for burns and complex surgical wound coverage with low risk of infection (23.4%), high template take rate (84% of patients had greater than 95% BTM take) and excellent autograft survival (92% of patients with greater than 95% STSG survival) in a 2025 meta-analysis of 202 patients [30]. There is also evidence supporting prompt application of BTM to debrided wounds and good outcomes when utilizing Meek Technique [27,30].

#### 3.3.3. Cultured Bilayered Skin Equivalents

Another major milestone in regenerative burn care has been the development of cultured bilayered skin, such as Epicel^®^ (cultured epidermal autografts) and OrCel^®^ (a bilayered cellular matrix containing neonatal fibroblasts and keratinocytes). These technologies are most applicable in patients with massive TBSA burns who lack sufficient donor sites for traditional grafting [31].

Epicel^®^, which uses autologous keratinocytes cultured over 2–3 weeks to generate large sheets of epidermis, has demonstrated utility in closing extensive wounds. However, its application is limited by fragility, delayed availability, and variable long-term durability. Integration with dermal scaffolds or synthetic matrices is often required to achieve biomechanical integrity [26,31].

More recently, 3D bioprinting and spray-on skin composites that incorporate autologous dermal fibroblasts, stem cells, and growth factors are being investigated as next-generation solutions to restore both dermis and epidermis in a single application. Preliminary results are promising but remain largely confined to early-phase trials and preclinical studies [32].

#### 3.3.4. Summary

A meta-analysis of reconstructive options in major burns (>50% TBSA) demonstrated highest graft take with autografts (96%) and lowest for cell-based therapy (72%); lowest number of operations with micrografting (4) and highest with cell-based therapy (9); shortest length of hospital stay with micrografting (50 days) and longest with cell-based therapy (91 days); and lowest mortality with cell-based therapy (11%) and highest with autografts (50%) [23]. The evolution of skin replacement strategies in burn care reflects a broader movement toward personalized, tissue-preserving, and functionally restorative interventions. Epidermal cell sprays, dermal regeneration templates, and cultured skin constructs are redefining the possibilities for wound closure in complex burns [25,26,31]. While limitations related to cost, availability, and clinical training persist, the cumulative evidence supports their integration into modern burn protocols, particularly when conventional autografting is suboptimal or infeasible. As these technologies continue to evolve—particularly in synergy with cellular and gene therapies—they hold promise to further improve outcomes in burn survivors across esthetic, functional, and psychosocial domains.

### 3.4. Energy and Metabolic Modulation

#### 3.4.1. Indirect Calorimetry and Metabolic Modulation

Severe burns induce a profound and sustained hypermetabolic response, marked by elevated resting energy expenditure, catabolism of lean body mass, and insulin resistance. This response may persist for over a year in large TBSA burns, contributing to delayed recovery, impaired wound healing, and increased morbidity [2,24].

While predictive formulas (e.g., the Curreri or Galveston formulae) have historically guided caloric supplementation, they often overestimate basal metabolic rates leading to overfeeding and adverse events [33].

Indirect calorimetry has emerged as the gold standard for determining actual energy expenditure in burn patients. By measuring oxygen consumption and carbon dioxide production, indirect calorimetry enables precise tailoring of nutritional support to avoid underfeeding (risking wound dehiscence and immunosuppression) or overfeeding (leading to hyperglycaemia, hepatic steatosis, and increased CO_2_ production) [33]. Over the past decade, portable and bedside indirect calorimeters have become more accessible, facilitating dynamic monitoring throughout the hospital stay.

There are limited clinical trials focused on severe burn injury rendering formulation of evidence-based guidelines for nutritional optimization challenging. There is limited literature on supplementation of omega-3 fatty acids, multivitamins (Vitamin A, B1 (thiamine), B6 (pyridoxine), B12, C (ascorbic acid), D and E (alpha tocopheryl acetate)), minerals (calcium and magnesium) and antioxidant micronutrients to reduce risk of wound infection and sepsis as well as reduce length of hospital stay [33]. The 2022 RE-ENERGIZE Trial of 1209 patients failed to demonstrate any reduction in time to discharge for patients with severe burns administered enteral glutamine without observation of any serious adverse events [34].

Both recombinant human growth hormone (rhGH) and insulin growth factor 1 (IGF-1) have demonstrated attenuation of hypermetabolism post burn, their side effect profiles not limited to hyperglycaemia and hypoglycaemia, respectively, have limited their clinical application. However, in severe burns, IGF-1 has been utilized in combination with insulin growth factor binding protein 3 (IGFBP-3) to minimize glycaemic fluctuation. The favourable cost and side effect profile of insulin, in addition to improved glycaemic control and enhanced wound healing make it a preferred option in hypermetabolism secondary to severe burns [10,33].

#### 3.4.2. Summary

Nutrition and modulation of the hypermetabolic response are critical components in the management of patients with burn injuries. Indirect Calorimetry is utilized to accurately determine each patient’s resting metabolic rate. Metabolic modulators such anabolic agents (GH, IGF-1, IGFBP-3, and insulin), catabolic antagonist propranolol, and anabolic steroid oxandrolone are employed in acute burn management [33].

### 3.5. Scar Modulation and Functional Recovery

Hypertrophic scarring remains a significant challenge in burn care, with functional and psychosocial consequences that can persist for years after wound closure. Scars may cause pruritus, pain, contractures, and disfigurement, particularly when they affect joints, facial areas, or other high-mobility regions [24,35]. Despite advances in acute burn management and wound closure, scarring continues to impact quality of life for many burn survivors. As such, scar modulation has become a critical focus of both clinical research and long-term care strategies. Over the past decade, fractional CO_2_ laser therapy has emerged as a transformative intervention in this space, supported by increasing high-quality evidence.

#### 3.5.1. Fractional CO_2_ Laser Therapy

Fractional carbon dioxide (CO_2_) laser therapy involves the delivery of microablative columns of laser energy into the dermis, promoting collagen remodelling and tissue regeneration while preserving surrounding skin. In burn scar management, it has been shown to improve scar texture, elasticity, pigmentation, and pliability, and to reduce associated symptoms such as pain and pruritus [8].

The laser achieves these effects through controlled thermal injury to fibrotic dermis, stimulating matrix metalloproteinases and subsequent collagen remodelling, while creating microscopic channels that enhance topical drug penetration. These biological effects mirror those of traditional scar revision surgery but with significantly less downtime and fewer complications [8,36].

Multiple studies, including prospective cohort studies and randomized controlled trials, have demonstrated the efficacy of fractional CO_2_ laser in improving both subjective and objective scar parameters [8,36]. More recently, a 2021 systematic review by Choi et al. concluded that fractional CO_2_ laser therapy is associated with statistically significant improvements in scar pliability, vascularity, pigmentation, and patient-reported outcomes [37].

Fractional laser therapy is now routinely incorporated into the comprehensive scar management protocols of leading burn centres globally. It is often initiated once re-epithelialization is complete, with sessions typically spaced 4–6 weeks apart. Adjunctive therapies—such as topical corticosteroids, 5-fluorouracil, or silicone gels—can be applied immediately post-laser to enhance outcomes [2,8,38].

The safety profile of fractional CO_2_ lasers in burn patients is favourable, with transient erythema, edema, and mild discomfort being the most commonly reported adverse effects. However, laser therapy requires clinician expertise, specialized equipment, and careful patient selection, particularly in darker skin types where post-inflammatory hyperpigmentation remains a concern [36].

#### 3.5.2. Rehabilitation and Functional Recovery

Burn rehabilitation is a multidisciplinary effort that begins in the acute phase and continues through long-term recovery. Early mobilization, physiotherapy, occupational therapy, and splinting play a central role in preventing contractures and optimizing function. As wound closure improves, attention shifts to scar flexibility, muscle strength, range of motion, and reintegration into daily activities [24,35].

Emerging evidence suggests that integrating fractional laser therapy with structured rehabilitation can yield synergistic benefits [8]. By improving scar pliability and reducing dermal tension, laser treatment facilitates improved joint mobility and reduces the mechanical limitations that often hinder physiotherapy. Several centres now use a combined protocol, alternating laser sessions with targeted physical therapy to maximize functional gains.

In addition, botulinum toxin is being investigated as an adjunct for scar modulation, particularly in the context of dynamic scars or those involving facial muscles. A growing body of literature suggests that botulinum toxin may reduce scar tension, inflammation, and fibroblast activity, although high-quality RCTs in burn-specific populations are still needed [39].

The long-term success of scar modulation also hinges on psychosocial rehabilitation. Burn scars often carry significant emotional and psychological burdens. Access to burn psychologists, support groups, and vocational rehabilitation services is critical in restoring confidence, body image, and social functioning. Patient-reported outcome measures (PROMs), such as the Brisbane Burn Scar Impact Profile or the POSAS (Patient and Observer Scar Assessment Scale), are now widely used in both clinical care and research to capture the full spectrum of burn recovery [40].

#### 3.5.3. Summary

Scar modulation is no longer viewed as an isolated post-burn concern but rather an integrated component of comprehensive burn care. Fractional CO_2_ laser therapy has revolutionized the management of hypertrophic scars, offering a minimally invasive, evidence-based approach to improve esthetic, functional, and symptomatic outcomes. When combined with tailored rehabilitation and psychosocial support, laser therapy contributes significantly to long-term quality of life and reintegration. As technologies evolve, the future of scar management will likely involve increasingly personalized protocols, leveraging genomics, biomarker profiling, and real-time imaging to optimize interventions for each patient [8,39,40].

### 3.6. Emerging Technologies and Future Directions

The evolving landscape of burn care is increasingly shaped by technological innovation and a shift toward individualized, precision-based medicine. While acute resuscitation and early wound closure remain priorities, growing attention is being paid to modulating systemic responses to injury, optimizing long-term outcomes, and leveraging data-driven tools to personalize therapy. Several key innovations—particularly in the domains of metabolic management, bioengineering, and digital health—are redefining how burn care is conceptualized and delivered [2].

#### 3.6.1. 3D Bioprinting and Tissue Engineering

A paradigm shift in skin reconstruction is underway with the advent of 3D bioprinting and tissue-engineered skin constructs. These technologies aim to recreate the structural complexity of native skin, including epidermal, dermal, and even vascular components, using a combination of cellular and acellular bioinks [31,32].

Early clinical trials have demonstrated feasibility in printing autologous skin substitutes directly onto wound beds, using patient-derived keratinocytes and fibroblasts embedded in collagen or fibrin matrices. The ability to control spatial deposition of cells and growth factors allows for tailored coverage of irregular wound geometries and functional zones. In preclinical models, bioprinted skin equivalents have shown promising rates of vascularisation, epithelialisation, and integration [28].

While these technologies are not yet widely available in clinical practice, significant investment in research and commercial development suggests that customized, autologous bioprinted grafts may become a viable option for extensive or functionally sensitive burn injuries within the next decade.

#### 3.6.2. Artificial Intelligence and Digital Burn Assessment

Artificial intelligence (AI), particularly machine learning algorithms, is beginning to play a role in burn depth assessment, wound monitoring, and predictive modelling. Tools incorporating computer vision have demonstrated accuracy comparable to experienced clinicians in classifying burn depth from digital images—a particularly valuable advance in remote or low-resource settings [41,42].

AI-driven algorithms have also been developed to predict complications such as sepsis, delayed wound healing, or graft failure, using real-time physiological data and laboratory markers. These tools may soon assist in triage, prognostication, and tailoring of treatment plans [2,41].

Moreover, telemedicine platforms enhanced with AI are improving access to expert burn care, allowing remote assessment and management in regions without specialist services. During the COVID-19 pandemic, several burn centres demonstrated the efficacy of teleburn clinics for outpatient follow-up, scar assessment, and rehabilitation.

#### 3.6.3. Precision Burn Care

Together, these innovations support the emergence of precision burn care—a framework that integrates genomics, metabolomics, imaging, and data analytics to tailor interventions to individual patient profiles [4,33,42]. Early studies are investigating the role of genetic polymorphisms in inflammatory and fibrotic responses to burns, with the potential to predict hypertrophic scarring or metabolic derangements.

Additionally, real-time biomarker monitoring of cytokines, lactate, and tissue oxygenation may soon enable adaptive resuscitation protocols, guiding fluid administration and vasopressor use more accurately than traditional parameters such as urine output or mean arterial pressure alone [4,33].

Precision care also extends into rehabilitation, with wearable sensors, app-based exercise programmes, and personalized scar modulation strategies becoming more commonplace. These tools empower patients and clinicians alike to track progress and intervene early when needed [42].

#### 3.6.4. Summary

Emerging technologies are rapidly transforming burn care from a reactive, protocol-based approach to a proactive, data-driven discipline. From indirect calorimetry and metabolic modulators to bioprinting, AI, and personalized rehabilitation, the next era of burn management will be defined by its capacity to anticipate, adapt, and tailor care to the unique needs of each patient [33,42]. The integration of these tools into routine practice, alongside robust clinical validation and cost-effectiveness studies, will be critical to ensuring equitable and widespread benefit.

## 4. Conclusions

Burn care has undergone a profound transformation, driven by a convergence of biologic insight, technological innovation, and multidisciplinary collaboration [2]. Contemporary burn management is no longer confined to the immediate goals of resuscitation and wound closure, but instead encompasses a continuum of care that prioritizes functional recovery, psychosocial reintegration, and long-term quality of life [24]. This shift reflects a deeper understanding of the complex systemic, cellular, and molecular responses triggered by burn injury and the recognition that optimal outcomes require a holistic, patient-centred approach.

A central theme in modern burn care is the adoption of evidence-based interventions that modulate the body’s response to injury. Pharmacologic agents such as propranolol and oxandrolone have emerged as powerful tools in attenuating the hypermetabolic state, reducing muscle catabolism, and accelerating wound healing [10]. These agents are increasingly incorporated into metabolic care protocols, particularly for patients with major burns, and highlight the growing role of targeted systemic therapy in what was once a surgically dominated specialty.

Advances in enzymatic and cell-based therapies have expanded the armamentarium for wound debridement and closure, offering less invasive and more selective alternatives to conventional surgical techniques [21]. Products such as bromelain-based enzymatic debriders, epidermal cell sprays, and bioengineered dermal substitutes allow clinicians to preserve viable tissue, reduce donor site morbidity, and promote better esthetic and functional outcomes [6,31]. The increasing availability of these technologies, along with improved protocols for their use, is narrowing the gap between laboratory innovation and clinical impact.

The management of hypertrophic scarring, a challenge in burn rehabilitation, has been aided by fractional CO_2_ laser therapy [8]. High-quality evidence now supports its use in improving scar pliability, pigmentation, and symptom burden, with favourable safety and tolerability. The integration of laser therapy into structured rehabilitation programmes exemplifies the modern multidisciplinary approach to burn recovery—where surgeons, therapists, and patients collaborate to restore function and form.

Looking ahead, emerging technologies such as 3D bioprinting, artificial intelligence, and telemedicine are poised to further personalize and democratize burn care [4,33,42]. These tools offer the potential to refine nutritional and fluid support, engineer patient-specific skin substitutes, improve diagnostic accuracy, and extend expert care to underserved regions. In parallel, advances in genomics, biomarker discovery, and digital health are ushering in an era of precision burn medicine, where care is guided not just by injury characteristics but by the individual’s unique biology and trajectory. Further studies should examine health economics and global implementation strategies to bridge gaps in burn care provision.

Yet, challenges remain. The widespread implementation of novel therapies is often limited by cost, availability, and the need for specialized training. Many promising interventions—particularly those in bioprinting and cell therapy—still require robust clinical validation and regulatory pathways for routine use [32]. Furthermore, the burden of burn injury continues to fall disproportionately on low- and middle-income countries, where access to even basic burn care remains limited. Ensuring equity in the application of these advances is a moral imperative for the global burn community.

In summary, the evidence-based management of burns has matured into a sophisticated, multidimensional field. The future of burn care will be defined not only by the development of new technologies, but by our ability to integrate them into comprehensive, individualized care pathways that optimize both survival and recovery. As research continues to unravel the biological, metabolic, and psychosocial complexities of burn injury, clinicians must remain agile—ready to adapt, innovate, and advocate for the patients they serve.

## Figures and Tables

**Figure 1 ebj-06-00059-f001:**
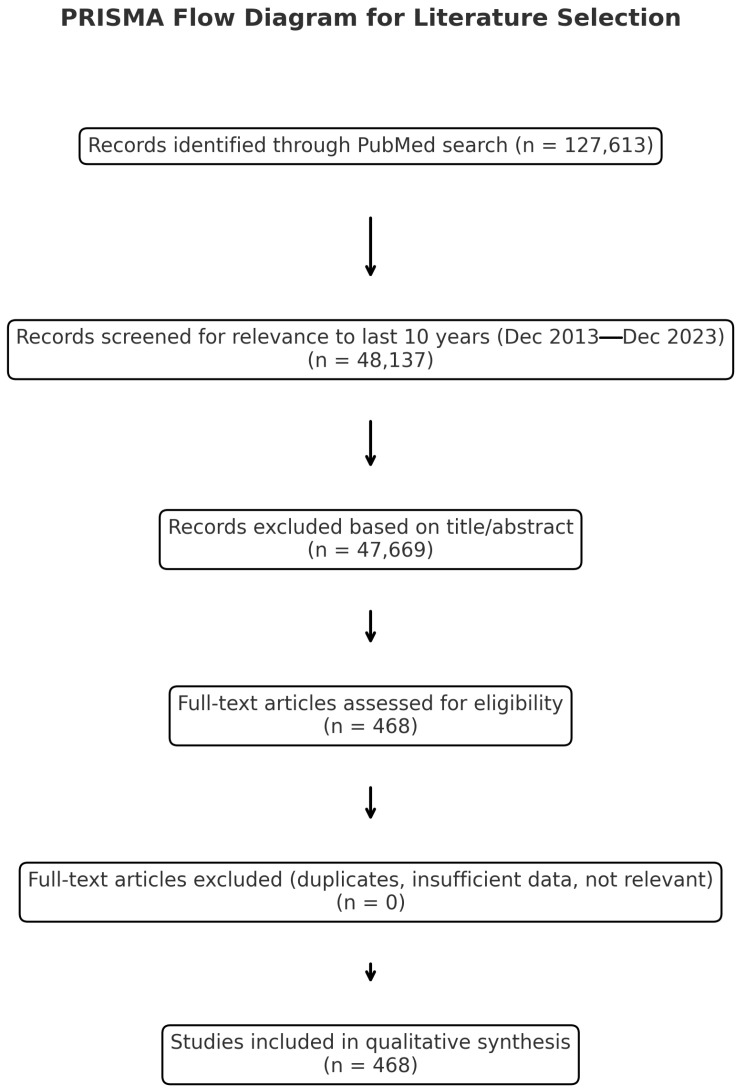
PRISMA flow chart.

## Data Availability

No new data were created or analyzed in this study. Data sharing is not applicable to this article.

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
