# Peer review of "Evidence-Based Management of Burns: A Narrative Review of Evolving Practices"

_2673-1991, 2025, doi:10.3390/ebj6040059_

Round 1
Reviewer 1 Report
Comments and Suggestions for Authors
Abstract and Introduction:
The abstract is clear and well-organized but slightly exceeds the typical journal length. It can be made more concise by removing minor redundancies and limiting brand names. The introduction provides strong context and rationale but repeats the theme of “individualized and evidence-based care” several times. Condensing those sentences would improve flow without losing depth.
Methods:
This section needs a bit more transparency. Although the authors describe using PubMed and reviewing approximately 48,000 articles with 468 finally included, it would strengthen the paper to briefly specify inclusion/exclusion criteria, the final date of the search, and how studies were categorized by topic. Even a short clarification would improve methodological credibility while keeping the narrative format appropriate for this type of review.
Discussion:
The review sections are detailed, well-referenced, and up to date. The pharmacologic adjuncts and debridement sections are particularly strong and well-supported by high-quality meta-analyses. However, there is some overlap between sections on metabolic modulation and pharmacologic therapy; consolidating or cross-referencing these may improve flow. The discussion on regenerative technologies is appropriate but could briefly mention cost-effectiveness or accessibility challenges, especially in low-resource settings.
Conclusion:
this section nicely integrates key findings and provides a balanced outlook. It could be strengthened by adding one or two sentences emphasizing future research priorities; such as multi-center randomized trials, health economics, or global implementation strategies to give a stronger forward-looking perspective.
References:
The reference list is comprehensive and current, which adds credibility. However, there appear to be duplicate entries (likely due to formatting artifacts at the end of the file). A careful check for duplication, numbering consistency, and uniform citation style is recommended.
Author Response
We thank Reviewer 1 for their time and effort in reviewing this manuscript. The contributions have elevated the quality of this manuscript. Please find below a point-by point discussion:
Comment 1:
Abstract and Introduction:
The abstract is clear and well-organized but slightly exceeds the typical journal length. It can be made more concise by removing minor redundancies and limiting brand names. The introduction provides strong context and rationale but repeats the theme of “individualized and evidence-based care” several times. Condensing those sentences would improve flow without losing depth.
Response 1: The abstract has been amended to meet the journal guideline of 200 words.
Comment 2
Methods:
This section needs a bit more transparency. Although the authors describe using PubMed and reviewing approximately 48,000 articles with 468 finally included, it would strengthen the paper to briefly specify inclusion/exclusion criteria, the final date of the search, and how studies were categorized by topic. Even a short clarification would improve methodological credibility while keeping the narrative format appropriate for this type of review.
Response 2: We have incorporated a short clarification as recommended by Reviewer 1 regarding the time frame of articles selected (page 2, 90) and added a PRISMA style flow chart (page 3, 105).
Comment 3:
Discussion:
The review sections are detailed, well-referenced, and up to date. The pharmacologic adjuncts and debridement sections are particularly strong and well-supported by high-quality meta-analyses. However, there is some overlap between sections on metabolic modulation and pharmacologic therapy; consolidating or cross-referencing these may improve flow. The discussion on regenerative technologies is appropriate but could briefly mention cost-effectiveness or accessibility challenges, especially in low-resource settings.
Response 3: We have amended the manuscript to incorporate address the overlap between sections (page 9, 383) on metabolic modulation and pharmacologic therapy to improve flow. In the summary section of regenerative technologies (page 8, 351), we have mentioned “While limitations related to cost, availability, and clinical training persist, the cumulative evidence supports their integration into modern burn protocols, particularly when conventional autografting is suboptimal or infeasible”.
Comment 4:
Conclusion:
this section nicely integrates key findings and provides a balanced outlook. It could be strengthened by adding one or two sentences emphasizing future research priorities; such as multi-center randomized trials, health economics, or global implementation strategies to give a stronger forward-looking perspective.
Response 4: We have included a sentence to emphasise future research priorities, “Further studies should examine health economics and global implementation strategies to bridge gaps in burn care provision” (page 13, 587).
Comment 5:
References:
The reference list is comprehensive and current, which adds credibility. However, there appear to be duplicate entries (likely due to formatting artifacts at the end of the file). A careful check for duplication, numbering consistency, and uniform citation style is recommended.
Response 5: We have carefully reviewed the references to amended for numbering consistency in keeping with journal guidelines.
Reviewer 2 Report
Comments and Suggestions for Authors
- A brief summary This is a paper looking at the current literature in burns management covering a wide range of topics.
- Comments regarding general concepts
Articles: It is very difficult to write a paper that is all emcompassing for burns - one would need to write several books on this. The journal has a very Australasian feel to it. For example, BTM, integra and Matriderm are described, but there are other products used widely in asia such as Nevelia. There are other enzymic products being used, though less effectively than Bromelin such as Iruxol.
Reviews: The topic is covered well, but by no means, is it able to cover the wide topic. The aim was to "synthesize high-quality evidence" but case reports were also included in the analysis and "Eligibility for inclusion/exclusion was assessed individually 99 by the senior authors, taking into account the aspects of novelty and relevance to the treat-100 ment of burns." Revising of the aim of the paper to reflect the content of the paper and how the papers were analysed would be prudent.
Author Response
We thank Reviewer 2 for their valuable comments
Comment 1:
A brief summary This is a paper looking at the current literature in burns management covering a wide range of topics.
Response 1: We thank reviewer 2
Comment 2:
Articles: It is very difficult to write a paper that is all emcompassing for burns - one would need to write several books on this. The journal has a very Australasian feel to it. For example, BTM, integra and Matriderm are described, but there are other products used widely in asia such as Nevelia. There are other enzymic products being used, though less effectively than Bromelin such as Iruxol.
Response 2: We thank Reviewer 2 in appreciating the complexity of summarising evidence based advances in burns literature into a single manuscript. We have attempted to provide examples of products with evidence base. In the absence of robust literature, the authors had to forego the inclusion of other possibly effective products such as Nevelia and Iruxol.
Comment 3:
Reviews: The topic is covered well, but by no means, is it able to cover the wide topic. The aim was to "synthesize high-quality evidence" but case reports were also included in the analysis and "Eligibility for inclusion/exclusion was assessed individually 99 by the senior authors, taking into account the aspects of novelty and relevance to the treat-100 ment of burns." Revising of the aim of the paper to reflect the content of the paper and how the papers were analysed would be prudent.
Response 3: We have incorporated a PRISMA flow chart to improve the clarity of how the papers were included in the analysis (page 3, 105).
Reviewer 3 Report
Comments and Suggestions for Authors
I prefer systematic reviews to narrative reviews because, unlike systematic reviews that follow standard guidelines such as PRISMA, narrative reviews do not have specific, well-known guidelines, which makes it challenging to review these studies. Anyway, reading this narrative systematics impressed me.
The following items are my opinions:
- The title clearly indicates the type of article (narrative review) and its subject
- The abstract summarizes the purpose, method, main findings, and conclusions.
- Keywords have been chosen correctly.
- I recommend the respected authors add similar to this explanation in introduction;
“Although the emergence of some kinds of burns management such as propranolol and oxandrolone for metabolic modulation; enzymatic debridement, regenerative approaches like epidermal cell sprays , dermal substitutes and innovations in scar modulation have passed more than 10 years ago, but these treatments have become more practical over the past ten years. “ or explain that “the history of using them in the treatment of burns is more than 10 years, but they have become more practical in the past 10 years”
- Method; why is search limited to PubMed? Wouldn't it be better to use also Scopus and Web of Science data resources to make a comprehensive understanding of contemporary, evidence-based approaches to burn care?
- Main body of the review; Although, the logical and clear organization and classification of the findings have been done, but using tables, graphs, and comparing findings could be very useful for summarizing data.
- It should be checked whether writing the trade names of some products in this manuscript is not considered commercial advertising?
To summarize, I am very impressed with findings of the current narrative reviews and for this reason, I believe to recommend acceptance with minor revision and I also would be happy to have the opportunity to review the revised version.
Author Response
The authors thank Reviewer 3 for the time and valuable comments that will no doubt elevate the quality of this narrative review.
Comment 1:
I prefer systematic reviews to narrative reviews because, unlike systematic reviews that follow standard guidelines such as PRISMA, narrative reviews do not have specific, well-known guidelines, which makes it challenging to review these studies. Anyway, reading this narrative systematics impressed me.
Response 1: We thank Reviewer 3 for the compliments provided regarding the quality of this narrative review
Comment 2:
The title clearly indicates the type of article (narrative review) and its subject
Response 2: Thank you
Comment 3:
The abstract summarizes the purpose, method, main findings, and conclusions.
Response 3: Thank you
Comment 4:
Keywords have been chosen correctly.
Response 4: Thank you
Comment 5:
I recommend the respected authors add similar to this explanation in introduction;
“Although the emergence of some kinds of burns management such as propranolol and oxandrolone for metabolic modulation; enzymatic debridement, regenerative approaches like epidermal cell sprays , dermal substitutes and innovations in scar modulation have passed more than 10 years ago, but these treatments have become more practical over the past ten years. “ or explain that “the history of using them in the treatment of burns is more than 10 years, but they have become more practical in the past 10 years”
Response 5: We have amended the introduction to include, “While some of these treatments have been around for longer, emerging evidence and practicality have resulted in more recent adoption into clinical practice (page 2, 83).”
Comment 6:
Method; why is search limited to PubMed? Wouldn't it be better to use also Scopus and Web of Science data resources to make a comprehensive understanding of contemporary, evidence-based approaches to burn care?
Response 6: We appreciate this valuable suggestion. Our choice of PubMed as the primary database was intentional, as it comprehensively indexes high-quality biomedical literature including MEDLINE, life sciences journals, and many surgical and reconstructive specialty publications relevant to burn care. While we acknowledge that Scopus and Web of Science could provide additional breadth, the overlap between these databases and PubMed for clinical and translational research in burns is substantial.
Comment 7:
Main body of the review; Although, the logical and clear organization and classification of the findings have been done, but using tables, graphs, and comparing findings could be very useful for summarizing data.
Response 7: We appreciate the reviewer’s constructive suggestion. We agree that visual summaries such as tables and graphs are valuable for data synthesis. However, this paper was designed as a narrative review rather than a systematic review or meta-analysis. Its purpose is to critically appraise and contextualize current trends in burn management rather than perform quantitative comparison of outcomes across studies. Given the heterogeneity of study designs, interventions, and outcome measures within the included literature, formal comparative tabulation or statistical synthesis would not be methodologically appropriate. Instead, we maintained a qualitative, narrative approach to highlight the most relevant evidence, supported by direct citations to recent systematic reviews and meta-analyses where quantitative data are already summarized.
Comment 8:
It should be checked whether writing the trade names of some products in this manuscript is not considered commercial advertising?
Response 8: All trade names mentioned in the manuscript (e.g., NexoBrid®, Integra®, MatriDerm®, RECELL®, NovoSorb® BTM) are cited solely for scientific clarity and reproducibility, not for promotional purposes. This approach aligns with European Burn Journal author guidelines and standard scientific writing practice for distinguishing between devices and formulations with distinct clinical evidence bases. We have reviewed the entire manuscript to ensure neutral phrasing and confirm that no statements imply endorsement or comparative superiority of any commercial product.
Comment 9:
To summarize, I am very impressed with findings of the current narrative reviews and for this reason, I believe to recommend acceptance with minor revision and I also would be happy to have the opportunity to review the revised version.
Response 9: We thank Reviewer 3 for their significant assistance in elevating the quality of this narrative review and look forward to any follow-up advice or recommendations.
Reviewer 4 Report
Comments and Suggestions for Authors
Dear authors,
Thank you for considering EBJ for the publication of your revised article titled "Evidence-Based Management of Burns: A Narrative Review of Evolving Practices".
Overall Assessment:
This review provides a comprehensive and timely overview of recent developments in burn management, combining advances in metabolic modulation, enzymatic debridement, regenerative technologies, scar modulation and emerging digital innovations. The paper transitions the discussion successfully from survival-focused care to holistic, patient-centred and technology-enhanced burn rehabilitation.
Major comments:
- The review is largely descriptive. Adding critical comparisons, such as limitations of key studies or areas of conflicting evidence, would enrich the analytical depth.
- The review could perhaps more explicitly address how cost and access barriers affect implementation in low- and middle-income countries.
Minor comments:
- While the methodology is outlined, the review would benefit from a PRISMA-style diagram or summary table detailing article selection, inclusion/exclusion criteria, and study types to enhance reproducibility.
- Some references appear redundant (e.g., duplicate numbering near the end of the reference list).
- Minor grammatical and typographical corrections would improve readability (e.g., “ad-vances” in the abstract).
We have no further comments.
Comments on the Quality of English LanguageGood.
Author Response
We thank Reviewer 4 for their expert advice and recommendations that will no doubt elevate the quality of this narrative review.
Comment 1:
Thank you for considering EBJ for the publication of your revised article titled "Evidence-Based Management of Burns: A Narrative Review of Evolving Practices".
Response 1: Thank you
Comment 2:
Overall Assessment:
This review provides a comprehensive and timely overview of recent developments in burn management, combining advances in metabolic modulation, enzymatic debridement, regenerative technologies, scar modulation and emerging digital innovations. The paper transitions the discussion successfully from survival-focused care to holistic, patient-centred and technology-enhanced burn rehabilitation.
Response 2: We thank Reviewer 4
Comment 3:
The review is largely descriptive. Adding critical comparisons, such as limitations of key studies or areas of conflicting evidence, would enrich the analytical depth.
Response 3: We thank the reviewer for this insightful comment. The manuscript was intentionally structured as a narrative review to summarize evolving concepts and evidence in burn management over the past decade.
Comment 4:
The review could perhaps more explicitly address how cost and access barriers affect implementation in low- and middle-income countries.
Response 4: Thank you for your helpful comment. We have attempted to address cost and access barriers—particularly as they pertain to implementation of novel therapies in low- and middle-income countries (LMICs). Specific examples and references from the manuscript include:
- Introduction: The review introduces burns as a significant burden in LMICs, explicitly noting that over 180,000 deaths from burn injuries occur “predominantly in low- and middle-income countries” (page 1, 40)and that “ongoing challenges include cost, access, and translation of novel therapies into widespread clinical practice.”
- Therapeutic Innovations: In the section on enzymatic (bromelain-based) debridement (page 6, 243), the manuscript describes how “cost and clinician familiarity remain barriers to widespread adoption, particularly outside tertiary burn centres.”​
- Regenerative Therapies: The review also highlights the limited use of cell-based therapies and dermal substitutes due to “cost, availability, and clinical training,” (page 8, 351) and notes that, “while limitations related to cost, availability, and clinical training persist, the cumulative evidence supports their integration into modern burn protocols, particularly when conventional autografting is suboptimal or infeasible.”​ (page 7, 294)
- Emerging Technologies: The manuscript indicates that, for next-generation treatments such as 3D bioprinting, “significant investment in research and commercial development” is required before they become widely available, and stresses the importance of “robust clinical validation and cost-effectiveness studies” for ensuring equitable benefits (page 12, 547).​
- In the Conclusions, it is reiterated that, “challenges remain. The widespread implementation of novel therapies is often limited by cost, availability, and the need for specialized training,” and that burn injuries disproportionately affect LMICs, “where access to even basic burn care remains limited (page 13, 593). Ensuring equity in the application of these advances is a moral imperative for the global burn community.”​
We have also amended, as per Reviewer 1, the conclusion, “Further studies should examine health economics and global implementation strategies to bridge gaps in burn care provision.” (page 13, 587)
Comment 5:
While the methodology is outlined, the review would benefit from a PRISMA-style diagram or summary table detailing article selection, inclusion/exclusion criteria, and study types to enhance reproducibility.
Response 5: We have incorporated a PRISMA style flow chart in methodology and amended the methodology to include time frames.
Comment 6:
Some references appear redundant (e.g., duplicate numbering near the end of the reference list).
Response 6: The references have been carefully reviewed and amended
Comment 7:
Minor grammatical and typographical corrections would improve readability (e.g., “ad-vances” in the abstract).
Response 7: We have reviewed the manuscript to amend minor grammatical errors and typographical corrections.
Round 2
Reviewer 2 Report
Comments and Suggestions for Authors
Thank you for the changes made
Reviewer 3 Report
Comments and Suggestions for Authors
Dear Sir
After reading the authors response, I have been convinced to recommend acceptance of the revised manuscript for publication
Reviewer 4 Report
Comments and Suggestions for Authors
Dear authors,
Thank you very much for considering EBJ for publication of your revised »Evidence-Based Management of Burns: A Narrative Review of Evolving Practices «.
You have considered our suggestions and made the necessary changes. We recommend a final proofread to correct any grammar mistakes.
We have no further comments.
Comments on the Quality of English LanguageGood.